# Cryo-EM reveals a phosphorylated R-domain envelops the NBD1 catalytic domain in an ABC transporter

Rodolpho Souza Amado de Carvalho*, Md Shamiul Islam Rasel*, Nitesh K Khandelwal, Thomas M Tomasiak

Many ATP-binding cassette transporters are regulated by phosphorylation on long and disordered loops which presents a challenge to visualize with structural methods. We have trapped an activated state of the regulatory domain (R-domain) of yeast cadmium factor 1 (Ycf1) by enzymatically enriching the phosphorylated state. A 3.23 Å cryo-EM structure reveals an R-domain structure with four phosphorylated residues and the position for the entire R-domain. The structure reveals key R-domain interactions including a bridging interaction between NBD1 and NBD2 and an interaction with the R-insertion, another regulatory region. We scanned these interactions by systematically replacing segments along the entire R-domain with scrambled combinations of alanine, glycine, and glutamine and probing function under cellular conditions that require the Ycf1 function. We find a close match with these interactions and interacting regions on our R-domain structure that points to the importance of most well-structured segments for function. We propose a model where the R-domain stabilizes a transport-competent state upon phosphorylation by enveloping NBD1 entirely.

## Introduction

ATP-binding cassette (ABC) transporters catalyze the translocation of a wide range of substrates across membranes and support a variety of physiological processes including ion homeostasis, adaptive immune responses, lipid distribution, and detoxification. They consist of two cytosolic nucleotide-binding domains (NBDs) that can bind and hydrolyze ATP which promotes conformational changes in their transmembrane domains (TMDs). Consequently, substrate binding and release are allowed in the transmembrane core driving transport [1, 2, 3, 4]. Mutations in ABC transporters cause numerous clinically relevant diseases, such as cystic fibrosis, cancer multidrug resistance, diabetes, atherosclerosis, coronary artery disease, and Tangier disease [5, 6, 7, 8, 9].

Eukaryotic ABC proteins are classified into seven subfamilies (A to G) based on their topology and domain packing, 5 of which are transporters (A, B, C, D, and G). The C-subfamily (called ABCC) is the largest ABC subfamily in the human genome and is characterized by a single polypeptide chain, containing two NBDs and three TMDs in most members (accessory TMD0 and common TMD1 and TMD2). The ABCC family also expands the canonical ABC architecture with the addition of a long disordered link between NBD1 and TMD2 called the regulatory domain (R-domain) in some ABCC transporters [4, 10, 11]. Some of the relevant transporters included in this family are CFTR, involved in chloride ion homeostasis; multidrug resistance protein 1 (MRP1) involved in the secretion of leukotrienes and drugs; the sulfonylurea receptor 1 (SUR1) a potassium channel that regulates insulin secretion; and the yeast cadmium factor 1, which is responsible for heavy metal and drug detoxification in yeast.

Several ABCC members present similarities in their regulatory core, especially the R-domain [12, 13]. Much of the foundational research of the R-domain extends from CFTR [14]. From the original cloning of CFTR, several canonical phosphorylation sites, mostly cAMP-dependent protein kinase (PKA), were discovered [15]. These were later shown in patch-clamp experiments to drastically stimulate CFTR transport [15, 16]. Further research identified additional kinases, including casein kinase II (CKII), protein kinase C (PKC), and cGMP-dependent protein kinase (cGK), as modulators of CFTR activity. CKII was explored through in vivo cellular assays, whereas PKC and cGK were examined using patch-clamp studies, illustrating the varied and significant roles these kinases play in regulating CFTR function [17, 18, 19]. In purified samples, mutagenesis on CFTR and Ycf1 can diminish ATPase activity and can impact cellular activity [20, 21]. Indeed, R-domain phospho-regulation seems to involve a complex kinase recruitment mechanism in which a myriad of phosphosites that suggest PKA, PKC, and CKII kinases can potentiate or suppress each other in the context of ABCC R-domain phosphorylation [22, 23]. Recent work points to a specific PKA site [24], S813, as the predominant rate-limiting site, similar to the Ycf1 site S908 [20, 25].

Department of Chemistry and Biochemistry, University of Arizona, Tucson, AZ, USA

Correspondence: tomasiak@arizona.edu
Nitesh K Khandelwal's present address is Department of Biochemistry and Biophysics, University of California – San Francisco, San Francisco, CA, USA
*Rodolpho Souza Amado de Carvalho and Md Shamiul Islam Rasel contributed equally to this work

The interactions of the R-domain with the rest of the transporter architecture have remained a mystery owing to its intrinsically disordered nature. Isolated fragments of the functional R-domain were identified as a random coil by circular dichroism (26). Early attempts to map specific R-domain regions important for activity initially relied on deletion studies (14, 26, 27). A patch-clamp study using CFTR R-domain deletions has shown that R-domain phosphorylation is required for the transport (26). Surprisingly, deletion of the entire R-domain results in only a minimal loss of activity in CFTR (14), with the C-terminal half being more important for PKA-dependent activity. A study with *Xenopus oocytes* expressing CFTR deletion mutants negatively impacted CFTR trafficking to the plasma membrane (27). Recent structures of the CFTR and more recently, Ycf1, have shown the phosphorylated R-domain binds to parts of NBD1 and intracellular loops (28, 29). Our previous triple-phosphorylated Ycf1 structure (29) showed this arrangement but only showed fragments of the R-domain and was missing the fourth phosphorylation site at S903. Notably, these R-domain positions lie directly over the CFTR F508 site (F713 in Ycf1). This site causes ~70% of cystic fibrosis disease-causing mutations and is found close to the R-insertion and R-domain intersection, which further highlights the importance of the R-domain regulatory environment (26).

More information is now being uncovered about the dephosphorylated state. Newer structures of the Ycf1 dephosphorylated state and the ABCC2 substrate-free state display that the R-domain also adopts an autoinhibitory structure in the substrate cavity, and CFTR has shown additional electron density between both NBDs (30, 31, 32, 33). The dephosphorylated Ycf1 structure showed this effect to be dependent on phosphorylation levels. Computational studies of low-energy conformations in the R-domain suggest that its phosphorylation-regulated intra- and inter-domain interactions might be mediated by molecular recognition elements (MoREs) (34, 35). These preformed elements promote interaction with disordered binding partners and enable rapid association and dissociation rates (36), which is in agreement with the later structural studies with Ycf1 and its dephosphorylated state (29, 32). Furthermore, these studies have shown that the R-domain itself adopts a more compact state, consistent with these observations (37). However, there have been no investigations comparing the stepwise de-activation from the substrate cavity or being able to image a complete interaction of the R-domain. Strikingly, there are no structures of any ABCC member available with the entire intact R-domain in any form, which is thought to be too mobile, to elucidate this mechanism. Although numerous CFTR structures are available in different states, once again they miss scanning the total R-domain sequence, which leaves a major gap in ABC transporter studies.

Here, we enriched the phosphorylated state of Ycf1 PKA sites using an enzymatic strategy used for CFTR to uncover a complete map of the R-domain. We determined the structure of this state to 3.23 Å resolution using cryo-EM and observed a new state phosphorylated in four positions along the R-domain with four overlapping phosphorylated motifs, PKC, PKA, and two CKII sites. We developed a new scrambling strategy to make scrambled chimeras in place of deletion constructs to maintain the length of the R-domain when ablating specific sites. We show that scrambling of those regions in context with our cryo-EM model impacts function to some extent with the most powerful effects near the phosphorylation sites. We also show that those regions not in contact do not impact cell survival. Taken together, this study provides a detailed description of the R-domain structure and critical phosphoregulatory interactions and reveals cellular effects that match closely to our novel structural observation.

## Results

### Cryo-EM structure determination of inward-facing wide PKA-phosphorylated Ycf1

We expressed and purified the NBD2 Walker B mutant E1435Q Ycf1 in the *Saccharomyces cerevisiae* DSY5 strain as described previously (29) and in the Materials and Methods section. After phosphorylation treatment with cAMP-PKA and size exclusion chromatography, cryo-EM grids were prepared as described in the Materials and Methods section. This approach yielded homogenous particles from which we identified a single inward-facing conformation in a 3.23 Å map (Fig 1A–C). The high-quality map allowed the first-time modeling of four phosphorylation sites (S903, S908, T911, and S914) in a single ABC transporter R-domain (Fig 1B and C). Because our previous structures of Ycf1 resulted in many cryo-EM classes (29) and a large amount of heterogeneity, we hypothesized that higher phosphorylation levels could drive Ycf1 into a uniform population. That way, we decided to further enrich the phosphorylated state by subjecting the purified Ycf1 sample to in vitro phosphorylation with PKA as described in a similar experimental setup to phosphorylated CFTR (28). In vitro phosphorylation yielded a relative increase in the overall phosphorylation level compared with WT as judged by the phospho-Q stain (Fig 1D), and the phosphorylation on S903, S908, T911, and S914 was confirmed with the mass spectrometry data (Table S1 and Fig S1A–C), consistent with the previous WT purified structure (29). When measuring the thermostability of different Ycf1 phospho-states, we observe an increase, about 5°C, in the melting temperature of our PKA-treated Ycf1 compared with the dephosphorylated Ycf1 (Fig 1E). The ATPase activity assay with different phospho-states of Ycf1 has shown that, upon PKA treatment, the ATP hydrolysis rate increases, whereas the dephosphorylated protein is reduced compared with the WT (Fig 2A). This suggests that phosphorylation modulates the regulation of this transporter which is consistent with previous studies with CFTR (31).

Besides that, the reconstruction yielded high-quality maps for most regions of the transporter and filled gaps from our previous structures (PDB: 7M69 and 7M68) (29), which allowed modeling of the amino gaps in TMD0 (125–132), TMD1 (329–332, 335–340), and NBD2 (1,259–1,267, 1,485–1,515). The overall conformation has previously been unobserved with a 40.0 Å separation between NBDs,

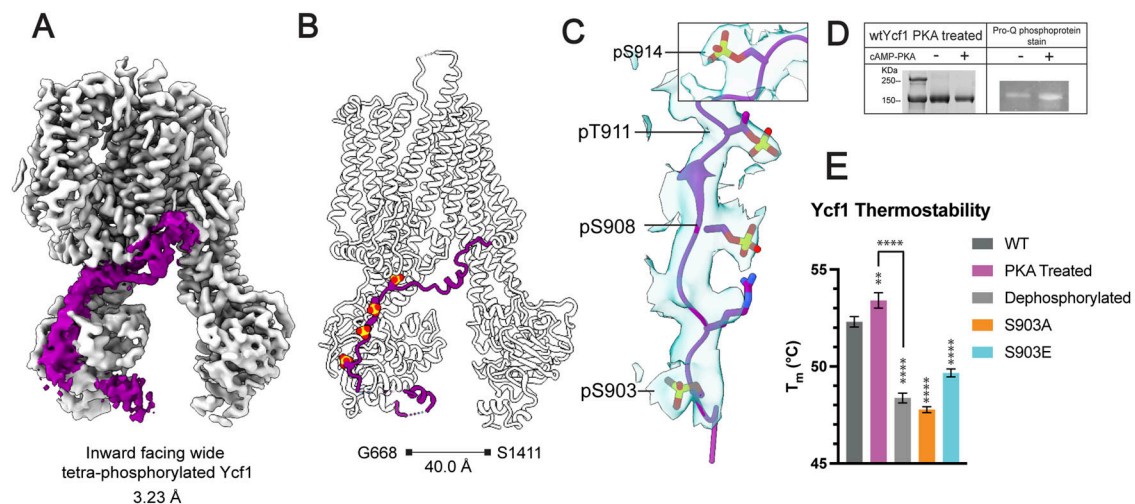

**Figure 1. Structure determination of the tetra-phosphorylated Ycf1 by cryo-EM.**
**(A)** The refined cryo-EM map of phosphorylated Ycf1 at 3.23 Å resolution. Ycf1 is presented as grey with the R-domain density shown in dark purple. **(B)** Model of phosphorylated Ycf1. The R-domain is shown in dark purple, with four phosphorylation sites visible in the overall R-domain placement. **(C)** A close-up of the cryo-EM density maps (RMSD 0.1205, 0.08 contour level, 2.5 selected radius) shows prominent phosphate groups associated with the residues S903, S908, T911, and S914. **(D)** The phospho-stained SDS–PAGE gel confirms the higher phosphorylation levels achieved with the PKA-treated Ycf1 sample compared with the WT protein. **(E)** Nano Differential Scanning Fluorimetry measurement presenting the melting temperature of the different phosphosites and S903 mutant. The data shown has been analyzed with one-way ANOVA and Dunnett's test that is representative of three biological replicates, **$P < 0.0021$, ****$P < 0.0001$.

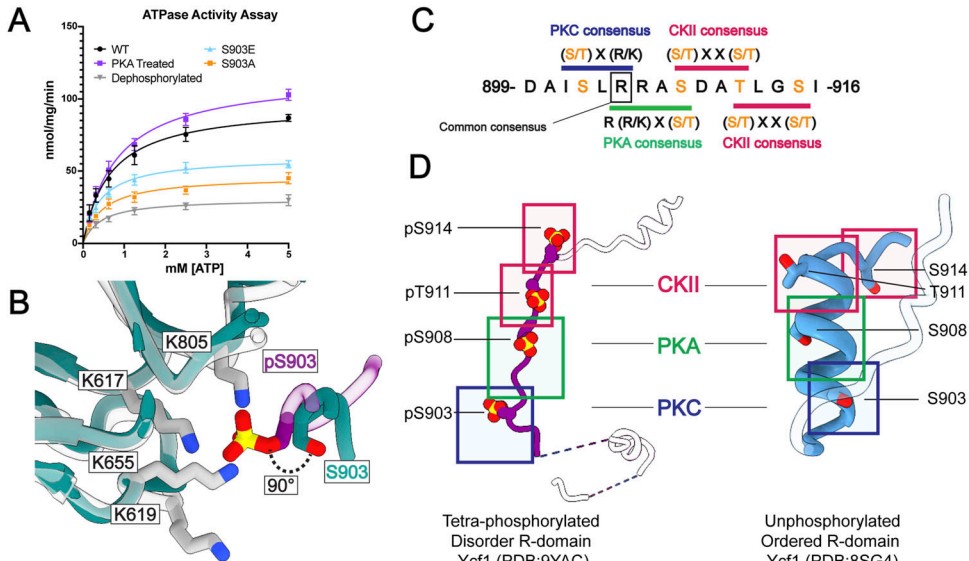

**Figure 2. Effects of phosphorylation in Ycf1 R-domain and overlapping kinase consensus motifs.**

**(A)** The different phospho-states of Ycf1 and the single-site mutant of S903 show relevant changes in the ATPase activity assay. **(B)** Comparison of the S903 site presented here (R-domain—purple) with the previous phosphorylated Ycf1 model missing the phosphorylation on S903 (colored green, PDB ID: 7M69, (1)), the serine residue is shifted at a 90° angle towards its binding pocket because of the presence of the phosphate. **(C)** The arrangement of phosphorylation sites in kinase motifs within the R-domain presenting a unique pattern. Overlapping kinase motifs suggest a complex phospho-regulation mechanism. **(D)** A comparison of the phosphorylated R-domain conformation to the unphosphorylated counterpart (2).

~5 Å wider separation than the previous widest model (G668 on NBD1 and S1411 on NBD2) (Fig 1B).

## The structure reveals an R-domain with four phosphorylation sites

The new structure reveals four total phosphorylation sites in the span of 11 residues, S903, S908, T911, and S914. Although we predicted S903 to be a PKC phosphorylation site by a computational prediction tool (Netphos 3.1) (38, 39) with a signature motif of (S/T) x (R/K), the phosphate occurrence at this site had not been

structurally confirmed. The new position of phosphorylated S903 shows a rotamer shifted 90 degrees towards NBD1 compared to the previous unphosphorylated models (Fig 2B). The S903 residue and its phosphate group make extensive interactions in a highly basic pocket defined by the surface of NBD1 (L619, K655, H803, K805, and K807) and R-insertion (615–643), especially at the K617 site, the latest being consistent with the R-insertion of CFTR in a proposed alternate conformation (40). In CFTR, although the R-domain is undefined, it still shares a similar basic binding pocket when compared with Ycf1 (Fig S2A–C). The ATPase activity assay conducted on the S903A and S903E mutants further underscores the

regulatory significance of this interaction site, demonstrating a significant decrease in the ATP hydrolysis rate of the mutants (Fig 2A).

This new arrangement sheds light on the peculiar arrangement of phosphorylation sites. From the confirmation of S903 phosphorylation in the Ycf1 sequence, we observe that the PKC-PKA-CKII-CKII sites (phosphorylated on S903-S908-T911-S914) share their kinase consensus sequences in an overlapping manner (Fig 2C and D). The individual PKC, PKA, and CKII recognition motifs are shared by at least one residue to the next kinase motif in the R-domain sequence: R905 shared by PKC and PKA motif; S908 shared by PKA and CKII and T911 shared between two CKII motifs (Fig 2C). This stabilized tetra-phosphorylated motif (PKC-PKA-CKII-CKII) makes a linear segment completely involved or bound to basic charges (Fig 2D). In contrast, in the previous dephosphorylated structure (32), they instead make a compact assembly in the 899–914 alpha helix that must be straightened to achieve this new conformation (Fig 2D).

### The phosphorylated R-domain completely wraps around NBD1

The cryo-EM maps of the enriched tetra-phosphorylated Ycf1 also reveal a continuous R-domain density (Fig 1A) that encloses entirely around NBD1 in a path resembling previously published Ycf1 structures (29, 41). The segments 882–891 and 900–935 were of enough quality to model and refine a polypeptide chain, but the rest of the R-domain (856–881, 892–899) density was too unfeatured to assign the correct sequence register to. Nevertheless, the rest of the density is continuous and reveals a general and individual domain topology that is in agreement with the predicted AlphaFold model (Figs S3A–D and S4A–C). We generated a second map using the local refinement procedure and particle subtraction in CryoSPARC masking out NBD2 to better isolate density around NBD1. In this locally refined map (3.4 Å), we could place a modified AlphaFold model with slight adjustments from our structure into this position to map, where the remaining R-domain interactions are likely to be (Fig S5).

The structure expands on our previous R-domain models (29) and traces its path along the periphery of NBD1 (Fig 3A and B). At its N-terminus, the R-domain coils to make a previously unobserved bridging contact with NBD2 (Fig S5A–C). The R-domain then returns towards NBD1 to make a helical structure (882–891) with important bridging contacts between negative charges along the R-domain (E885, D893, D895, and D900) to a series of histidines on NBD1 (H786, H790, H794, H803) (Fig S5D). After this segment, the R-domain's four phosphorylated residues extend the electronegative segment with extensive contacts to a positively charged groove along the TMD2-NBD1 surface (Fig 3C). Direct contacts amenable to ionic interaction are observed among three of the six residues in this cluster: the positively charged residues K617, K655, and K805 at 3.3, 3.4, and 2.5 Å respective distances to the phosphate oxygen group (Fig 3E). The other three residues K619, H803, and K807, although further apart from the phosphate (~7–8 Å) might be weakly contributing through long-range electrostatic interactions. The phosphorylated residues S908, T911, and S914 presented a binding pattern similar to what has been observed previously (Fig 3D and F). Finally, the R-domain ends at the elbow helix of TMD2 with a short helical segment as described before.

Finally, the patterning of charged and hydrophobic interactions along the entirety of NBD1 reveals a common feature in the bound state of intrinsically disordered domains (42, 43). The charged regions of the R-domain are intercalated with hydrophobic contacts. The pairs of R-domain S908-T911-L912-I915 with NBD1 T1145-F1151-I1154, as S914-F917 and I1123 were observed in the previous Ycf1 structure. We now infer the R-domain positions of S903-A901-I902-L904 contacting NBD1 I621-F721-L801-L802.

### Cellular evaluation of R-domain contacts through the linker replacement strategy

To assess the importance of the R-domain, we systematically altered segments of the R-domain in a linker replacement strategy by mutation of 6-10 amino acid segments to poly glycine-asparagine-alanine. The rationale was twofold. First, we reasoned that Ycf1 R-domain is shorter than the CFTR R-domain and that deletion constructs previously used in CFTR (14, 26, 27) would interfere with proper folding in Ycf1. Second, we reasoned that polar and flexible constructs would be the least intrusive replacement when still measuring the loss of specific contacts. The linker insertion mutants created were 855–864, 865–874, 875–884, 885–894, 895–904, 905–912, 913–919, 920–926, and 927–935, numbered as the residues of the segment scrambled (Fig 4).

The effect of each insertion was tested on a cadmium survival assay (100 $\mu$M CdCl$_2$) in a Ycf1 knockout strain dependent on mutant Ycf1 function for survival. We also assessed the full R-domain scrambled construct and as controls, we evaluated the E1435Q catalytic dead mutant and the empty vector that is missing the *Ycf1* gene as shown in previous studies (21, 29, 32). The R-domain scrambled mutants generally showed a significant reduction in survival compared with the WT, with the strongest effects localized to the regions near the phosphorylation sites (segment 895–904, 905–912) and the C-terminus (927–935) (Fig 4A–C). All other regions in contact with NBD1 in the cryo-EM structure showed defects in growth although to a lesser degree. Unexpectedly and in contrast to CFTR deletion experiments (14), the full R-domain scrambled construct had a severe effect. The general pattern exactly matches the cryo-EM structure, with regions in contact between the R-domain and NBD1 being necessary for survival and those that we observe as not in contact are not important (Fig 4B). This trend overall matches that of the AlphaFold model pLDDTT confidence scores, with higher confidence scores being associated with higher requirements for survival.

Strikingly, the segments 884–894 and 927–935, which do not contain phosphorylation sites, had a drastic impact on survival similar to the key phosphorylated sites mentioned earlier (Fig 4A). Thus, the 884–894 region matches closely with the novel R-domain alpha-helix (882–891) modeled in our structure, so we hypothesize that this hydrophobic and charged helix might be generating an important, novel, R-domain interaction with implications in Ycf1 activity. In terms of the 927–935 scrambled importance, our structure also reveals potential charge-charge

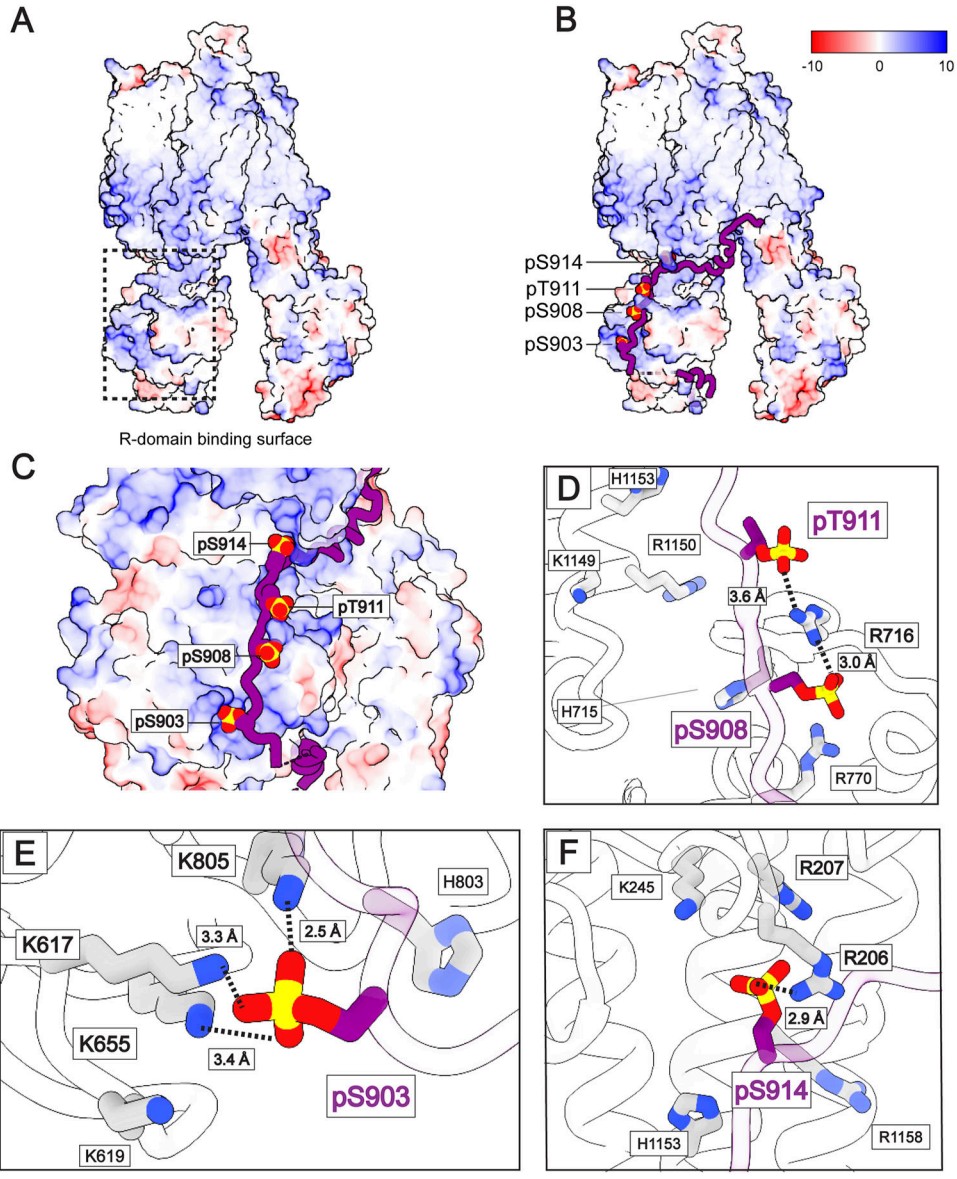

**Figure 3. R-domain engagement with NBD1.**
**(A)** The electrostatic potential surface of Ycf1 (min −24.36, mean 1.23, max 32.50 kcal/(mole)) with the R-domain density removed to illustrate the strong basic potential of the NBD1 surface (dashed box). **(A, B)** The tetra-phosphorylated R-domain is closely represented in cartoon form on the rest of the Ycf1 surface colored as in (A). **(C, D)** The enclosed view of the interacting basic surface zone (D). The S908 and T911 phosphate groups are highlighted with their direct ionic interaction pair, the R716 in NBD1 with similar contacts as in (1). Also highlighted are other basic residues in the proximal regions adding to the overall charge in this micro-environment, such as R770, R1149, and R1150. **(E)** The phosphorylated S903 site with the respective contacts to K617, K619, K655, H803, and K805. **(F)** S914 residue is encircled by the positively charged residue R206 as shown in (29). In all subfigures, the R-domain is colored purple, the rest of Ycf1 is colored grey, nitrogens are blue, oxygens are red, and phosphorous is colored yellow.

interaction between the helix and a positively charged region (R1112, R1115, R1119).

Comparatively, the scrambled mutants presented a similar survival trend to what has been observed with single-point mutants (S903A, S908A, T911A, or S914A) in previous studies (21, 25, 29, 32). The scrambled 895-904 and S903A had a similar drastic effect on cellular survival. Interestingly, the S903E mutant did not present phospho-mimetic activity as observed in previous studies with S908D/E and T911D and instead led to a loss of function (21). The glutamic acid substitution in S903 may not be sufficient to recall the ionic interaction from the phosphate group, which is extensive in the phosphorylated S903 structure. As mentioned earlier, the S903 binding pocket recruits three immediate ionic partners instead of one, as observed in S908 and T911, that way the single charge provided by the glutamic acid cannot replace the contacts of phosphorylated S903.

**The R-domain contains co-evolving kinase motifs in the phosphoregulatory region**

In our previous work, we performed an evolutionary coupling analysis using EV couplings (44) to understand how the R-domain regions coupled to each other and the rest of the Ycf1 (29). Here we reperform the analysis and focus on how segments of the R-domain co-evolve with each other in light of the interactions that are observed. The R-domain generally reveals few contacts because of the high variability in this region. We find that the entire segment of continuous phosphorylation sites from residue 905 to 914 co-evolve as one linear segment (Fig S6A–C). This segment then co-evolves with another segment, 872–875 on the R-domain N-terminus. Interestingly, this site also contains numerous presumptive phosphorylation motifs of PKA, PKC, and CKII arranged in a similar order to the functionally important site in S903-S914. These sites are

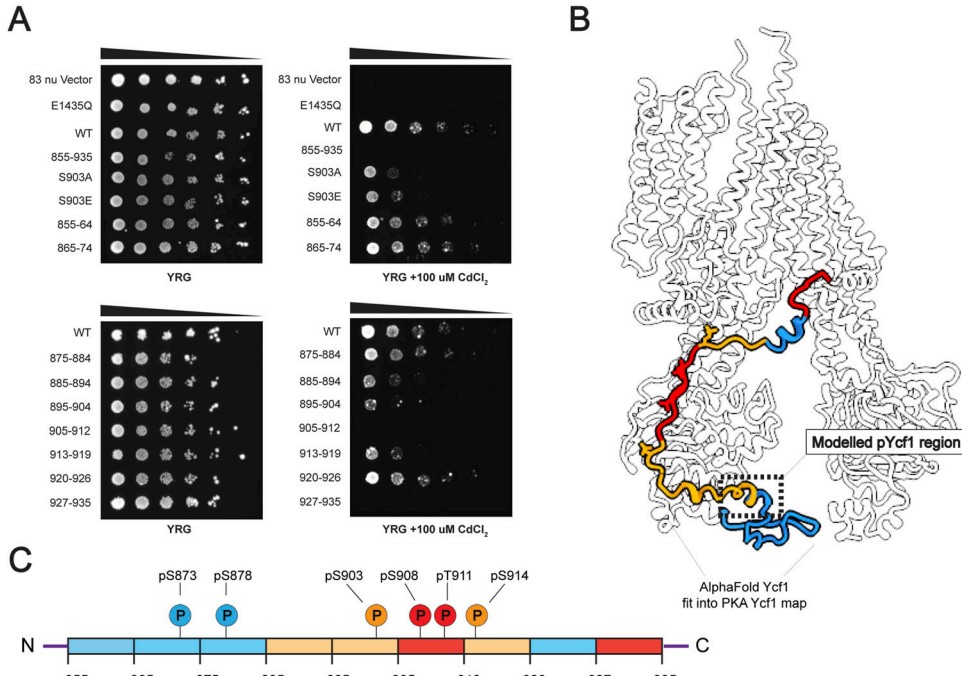

**Figure 4. Yeast survival assay with scrambled R-domain constructs.**
**(A)** *S. cerevisiae* strain BY4742 with endogenous Ycf1 knocked out expressing different Ycf1 scrambled constructs for the highlighted R-domain segments. Cellular growth is serial diluted (1:5) and plated in a control YRG and CdCl2 toxic YRG plates. **(B)** A representation of the linear sequence of Ycf1 R-domain with respective phosphorylation sites. **(C)** The R-domain scrambled mutants colored by impact to survival. Red regions showed the most severe defects, blue regions showed no defect, and yellow regions were intermediate. The data shown are representative of four biological replicates.

phosphorylated in our data here and previously published data. Strikingly, phosphorylation of S878 seems not to impact cellular transport, as observed in our scrambled mutants and previous studies have also shown no impact on ATPase activity (32).

## Discussion

The Ycf1 structure in this study shows an enriched PKA-phosphorylated state that unveils new features within the ABCC family transporters which explain the interaction of the phosphorylated R-domain. It describes a fourth phosphorylated site, S903 which has not been structurally observed before, co-evolves as a critical unit with S908, T911, and S914 and for the first time provides cryo-EM density that shows the entire path of the R-domain. A new scrambled linker insertion functional assay shows in an unbiased way that this segment is the predominately important region for survival on cadmium, with supporting regions that flank it. Importantly, the cellular assay mirrors the regions of the cryo-EM maps that can be visualized with R-domain density in direct contact with NBD1. This supports a model where R-domain phosphorylation predominantly stabilizes in the NBD1 through contacts around the periphery of NBD1 (895–919), whereas a major role and most severe defects are localized to 905–912. This exactly coincides with the pLDDT scores in the AlphaFold model of Ycf1.

This structure enables the first modeling of an intact regulatory domain presenting four phosphates occupying their respective binding sites. Density is visible for the entirety of the R-domain (855–935) and matches closely with the AlphaFold prediction of this region, whereas detailed atomic modeling was possible in residues

882–891 and 900–935. Taken together, the secondary structure assignment also closely matches the proposed one for CFTR in an analogous R-domain region of the phosphorylated state found by NMR (45). Previously, two independent phosphorylated Ycf1 structures (7M69 and 7MPE) (29, 41) and the structure of CFTR (45) showed that the NBD1 supports binding of the R-domain, here the extended R-domain regions present a similar morphology further enveloping the NBD1. Comparatively, these phosphorylated structures resemble the overall global conformation of the dephosphorylated auto-inhibited Ycf1 structure (32), except for the R-domain placement. This phosphorylated Ycf1 structure and our previous models suggest that disordered regions can acquire an array of R-domain conformations with distinct structural characteristics displaying regulatory function.

The phosphorylated residue (S903) shown here can directly interact with a highly basic pocket in the lower NBD1 portion and drive partial Ycf1 function in our cellular assay. The crucial role of serine phosphorylation in the R-domain in CFTR activation has been extensively described (15, 46, 47). Similar studies have suggested S908 phosphorylation as the rate-limiting step for Ycf1 activity (32). Here, we demonstrate how the requirement for S903 phosphorylation is also linked to transport activation. Nonetheless, CFTR studies have shown that combined PKA and PKC stimulation can drastically increase R-domain complex formation and activation (23, 48, 49). The phosphorylated S903 site phosphate engages a distinct triple ionic interaction with the basic groove in the N-terminal region (K617, K655, and K805), which stands out when compared with the other phosphosites (S908, T911, and S914). Critically, the interaction between the K617 site and S903 involves a region that is homologous to another vital regulatory area in CFTR, known as the R-insertion (28), suggesting an R-domain and

R-insertion interaction mechanism. In CFTR, this interaction is likely to be a proposed alternate state of the R-insertion that brings it into contact with the R-domain but is present natively in Ycf1 (40). These interactions likely stabilize the phosphorylated R-domain outside of the TMD to sequester it during the transport cycle. The recent structures of human Mrp2 with different conformational states suggest a similar mechanism in which the auto-inhibited state must be disengaged upon substrate binding (30, 50, 51 Preprint).

Finally, this structure also reveals broad physical-chemical properties important for disordered domains (49). The R-domain is dominated by alternating charged regions from the phosphorylation sites and then aspartates/glutamates interspersed by hydrophobic regions. This feature is suggested to favor binding towards hydrophobic grooves and promote stabilization less stringently (42, 43, 49). This type of dynamic interface is proposed to be advantageous to conformational fluctuations and facilitate post-translational modification in a signaling network (43, 52, 53, 54, 55). Similarly, the hydrophobic lower NBD1 interactions with the R-domain (Fig S5) supported by our 884–894 scrambled mutant indicates that non-phosphorylated networks can also impact activity. This N-terminus segment, although not conserved, still provides information regarding the low enthalpy state that the R-domain needs to achieve in this inward-facing conformation.

In conclusion, our tetra-phosphorylated Ycf1 structure contributes to a detailed understanding of the post-translational modification required for regulation in a relevant group of transporters, specifically the ABCC family. The structural elucidation of a novel phosphorylated residue with a well-defined binding site confirms the existence of additional regulatory sites with different mechanisms in the same regulatory core. This discovery may have implications for numerous other transporter classes featuring similarly unstructured domains or loops with multiple phosphorylation modes. These findings are also in agreement with a current CFTR structural study demonstrating kinase activation (56 Preprint). Furthermore, this precisely identified regulatory region has the potential to serve as a foundation for a complex activation mechanism and allosteric modulation in various cellular contexts.

# Materials and Methods

### Cloning, expression, and purification

The *S. cerevisiae* YCF1 (yeast cadmium factor 1) gene was codon-optimized and cloned into the p423_GAL1 yeast expression vector as an N-terminal Flag (DYKDDDDK) and C-terminal deca-histidine (10X His) tagged fusion protein (GenScript) (Fig S7A). The E1435Q, R-domain phosphorylation sites and interacting residues mutants were generated by site-directed mutagenesis using a Q5-New England Biolabs PCR mutagenesis kit together with primers synthesized from Millipore Sigma-Aldrich and verified by sequencing (Elim Biopharmaceuticals, Inc.). For protein expression, the *S. cerevisiae* strain DSY-5 derivative (Genotype *MATα his3::GAL1-GAL4 pep4 prb1-1122*) was transformed with the Ycf1 expression plasmid construct and grown in YNB-HIS selection media plates (57). Cellular growth was inoculated into a 50 ml primary culture grown for

at least 24 h at 30°C using a ThermoFisher MaxQ 8000 incubated stackable shaker (SHKE8000) with shaking at 200 rpm in YNB-His media (0.67% wt/vol yeast nitrogen base without amino acids, 2% wt/vol glucose, and 0.08% wt/vol amino acid dropout mix without histidine). A secondary 750 ml culture of YNB-His media was inoculated with 2% of the primary culture (15 ml) and grown under the same growth conditions for an additional 24 h before induction by adding YPG media (1% wt/vol yeast extract, 1.5% wt/vol peptone, and 2% wt/vol galactose final concentration) from a 4X YPG media stock. The culture was grown for an additional 16–18 h at 30°C before harvesting by centrifugation at 4,000*g* for 45 min at 4°C.

For protein purification, harvested cells were resuspended with ice-cold lysis buffer (50 mM Tris–Cl pH 8.0, 300 mM NaCl, and complete, EDTA-free protease inhibitor cocktail tablets [Roche]) at a ratio of 3.1 ml/g of cell pellet. Resuspended cells were lysed on ice by bead beating with 0.5 mm glass beads for eight cycles consisting of 45 s of beating, with 5 min between cycles. Lysates were collected by vacuum filtration through a coffee filter and membranes were harvested by ultracentrifugation at 112,967*g* for 1.5 h before storage at −80°C. Membranes were solubilized in resuspension buffer (50 mM Tris–Cl pH 7.0, 300 mM NaCl, 0.5% 2,2-didecylpropane-1,3-bis-$\beta$-D-maltopyranoside [LMNG]/0.05% cholesteryl hemisuccinate [CHS] supplemented with protease inhibitor as described above) at a ratio of 15 ml/g of membrane at 4°C for 4 h. Solubilized membranes were clarified by centrifugation at 34,155*g* for 30 min at 4°C. The clarified supernatant was filtered through a 0.4 $\mu$m filter to remove the insoluble fraction and supplemented with 30 mM Imidazole pH 7.0 immediately before loading at a flow rate of 2 ml/min onto a 5 ml Ni-NTA immobilized metal affinity chromatography column (Bio-Rad) equilibrated in Buffer A (50 mM Tris–Cl, 300 mM NaCl, 0.01% LMNG/0.001% CHS, pH 7.0). After loading, the column was washed with 10 column volumes (CV) of Buffer A to remove nonspecifically bound proteins then followed by a gradient of Buffer B (50 mM Tris–Cl, 300 mM NaCl, 500 mM Imidazole 0.01% LMNG/0.001% CHS, pH 7.0) consisting of the following step sizes: 6% (10 CV), 10% (2 CV), 16% (2 CV), and 24% (2 CV). Protein was eluted with 4 CV of 60% buffer B and immediately diluted 10-fold with Buffer A before concentration and three rounds of buffer exchange to remove excess imidazole by centrifugation at 3,095*g* at 4°C in 100 kD cutoff concentrators (Amicon). Concentrated, buffer exchanged sample was lastly purified by size exclusion chromatography (SEC) at 4°C by injecting sample onto a Superose 6 Increase 10/300 GL column (GE Healthcare) equilibrated in SEC buffer (50 mM Tris, 300 mM NaCl, pH 7.0) supplemented with either 0.01% LMNG/0.001% CHS (Fig S7B and C) or 0.06% glyco-diosgenin (GDN) (Fig S7D and E) and immediately used for biochemical assay or cryo-EM grid preparation following quantification by BCA Assay (Pierce).

### Ycf1 phosphorylation and dephosphorylation reactions

After size exclusion purification, both WT and E1435Q YCF1 were concentrated in a 100 kD cutoff concentrator (Amicon) and BSA quantification was performed. For phosphorylation reactions, the concentrated protein at 3 mg/ml was incubated with cAMP Protein Kinase A PKA (NEB) (molar ratio of 40:1) at room temperature for 1 h. A total reaction mixture of 500 $\mu$l was used for downstream size exclusion purification using a superose 6 10/300 column (GE

Healthcare) to remove excess PKA and phosphorylation reagents (Fig S8). Dephosphorylation reactions were performed with lambda Protein Phosphatase (lambda PP) (NEB). 1,000 units of lambda PP were used per $\mu$mol of Ycf1 protein in a 500 $\mu$l reaction. After a 1-h treatment at 30°C, further size exclusion chromatography was performed in a Superose 6 10/300 column (GE Healthcare) for the removal of dephosphorylation reagents.

## Cryo-EM sample preparation and data acquisition

Cryo-EM grids for E1435Q PKA-treated Ycf1 were prepared immediately following a second SEC purification for E1435Q Ycf1 protein after treatment with protein kinase A (PKA) (Fig S7E). 5 $\mu$l of concentrated E1435Q (5.94 mg/ml) PKA-treated Ycf1 sample was applied to a QF-1.2/1.3-4Au 400 mesh grid (E1435Q Ycf1) purchased from Electron Microscopy Sciences. Grids were placed inside a Leica EM GP2 equilibrated to 10°C and 80% humidity. After a 10-s incubation, the side of the grid to which the sample was applied was blotted on Whatman 1 paper (8 s for WT; 3.5 s for E1435Q), then immediately plunged frozen in liquid ethane equilibrated to –185°C. A total of 8,587 movies were captured for E1435Q, on a Titan Krios at 300 kV equipped with a K3 Summit detector (Gatan) at the Pacific Northwest Center for Cryo-EM. Movies were collected at 22,500X magnification with automated super-resolution mode and defocus ranges of –0.9 to –2.1 $\mu$m. Movie frames contained 60 frames with a per frame exposure of 0.9 electrons/$Å^2$ dose rate (~54 electrons/$Å^2$ total dose).

## Cryo-EM data processing

The phosphorylated Ycf1 E1435Q dataset was processed in CryoSPARC (version 4.2.1) (58). 8,587 micrographs were motion corrected by CryoSPARC Patch Motion Correction and drift correction to generate an image stack with a pixel size of 0.822 Å/pixel. The contrast transfer function (CTF) was estimated for dose-weighted micrographs using CryoSPARC Patch CTF before particle picking using the automated Blob picker. Interactive selection of particles was performed with Inspect Particle Picks on the total automated picks and subject to reference-free 2D Classification to generate references. Selected 2D classes representing the Ycf1 expected morphology were used for a reference-based particle picking with the Template picker function. 2D Classification was once again performed and the best classes were used on ab initio reconstruction. Several rounds of 3D heterogeneous and non-uniform refinements in CryoSPARC (58, 59) were performed leading to a 3.89 Å resolution map of phosphorylated Ycf1 obtained (Fig S9). This map was then used as a reference for artificial intelligence-assisted TOPAZ (60, 61) particle picking in CryoSPARC in the initial micrograph dataset (Fig S10). A total of 2,159,582 particles were automatically picked and extracted with 4X binning resulting in a box size of 440 pixels with 6.576 Å/pixel. Multiple rounds of 2D classification were performed to remove bad particles resulting in 1,626,297 particles subject to 3D analysis in ChimeraX after extraction with 2X binning and a box size of 220 pixels with 2.062 Å/pixel.

Iterative rounds of local and global CTF refinements in association with heterogeneous and non-uniform refinement were used to achieve a final 3.23 Å map, using 68,169 particles. Local refinement was performed in the final map using particle subtraction, in which NBD2 was masked out to decrease particle heterogeneity from this dynamic domain. Local refinement generated a 3.4 Å map with the expanded region of the N-termini R-domain portion in association with NBD1. CryoSPARC 3D flexible refinement was also performed using the 3.23 Å map and particle stack to generate a 3D volume series that represents conformational motion in the phosphorylated sample. Local resolution and FSC validation were performed in CryoSPARC. The final and locally refined map was then used for manual model building as specified below. Data-processing workflow is shown in Figs S9 and S10, the final EM maps, and the quality report.

## Model building and refinement

An initial model of Ycf1 was built using the ISOLDE suite (version 1.3) in UCSF ChimeraX (version 1.3) (62) and the AlphaFold2 (63, 64) Ycf1 model was used as the initial reference template. Manual model building was performed in both COOT (version 0.9) and ISOLDE ChimeraX (65, 66). Multiple Iterative cycles of real-space refinement and analysis were performed in Phenix (version 1.20.1) and CCP-EM modules (56 Preprint, 58). Secondary structure restraints were extensively used as an additional Phenix restrain. For the structure-building process map blurring and sharpening features of the COOT package were used for structure analysis, especially on locally refined maps for R-domain modeling. Molprobity dedicated web service was also used in association with the Phenix platform to optimize geometry. To maintain proper geometry, starting model restraints and harmonic restraints were used extensively in Phenix. The final model refinement statistics are shown in Table S2. Model visualization, analysis, and figure preparation were performed using UCSF ChimeraX.

## ATPase activity assay

For evaluating ATPase activity, WT and mutants were expressed and purified (Fig S8) as described above in a buffer containing 0.01% LMNG and 0.001% CHS. Different phosphorylation states were achieved by enzymatic treatments (cAMP PKA and lambda PP phosphatase) as described above, and a second size exclusion was performed to exclude the enzymatic treatment components, after BCA quantification. The ATPase rates were measured as a colorimetric endpoint assay for inorganic phosphate detection described by reference 67, which allowed the indirect quantification of protein-driven ATP hydrolysis. The Michaelis-Menten kinetics for the ATPase rates of three biological replicates at an ATP range (0.05–4 mM) are reported as the mean ± standard quantified in GraphPad Prism. For each biological replicate, two technical replicates were performed. Reaction was performed near the physiological growth temperature (30°C) of its host organism, *S. cerevisiae*, for 30 min. The ATPase reaction was started by the addition of an Mg-ATP solution prepared in Tris buffer, bringing the final concentration of Ycf1 to 0.135 mM in a 50 $\mu$l reaction containing 10 mM $MgCl_2$ and varying final concentrations of ATP (0–4 mM). Samples in the absence of Mg-ATP solution or Ycf1 were prepared as negative controls. The reaction was quenched for a single time point by adding 40 $\mu$l of 5% SDS. After quenching, 200 $\mu$l of detection reagent (8.75 mM ammonium molybdate, 3.75 mM zinc acetate, pH

5.0, and 7.5% ascorbic acid, pH 5.0, prepared fresh before use) was added to each sample and incubated for 25 min at 37°C for revelation. Data were fit using nonlinear regression in GraphPad Prism 9 to derive Vmax values.

### NanoDSF thermostability measurements

All samples were run on a NanoTemper Tycho NT.6 nanoDSF instrument (NanoTemper Technologies). Samples were set up in triplicate at a final concentration of 0.45 mg/ml in NanoTemper capillaries using regular purification buffer (50 mM Tris–Cl, 300 mM NaCl, 0.01% LMNG/0.001% CHS, pH 7.0). All Ycf1 samples were measured at a scan rate of 30°C per minute over a temperature range of 35–95°C. Samples were excited for the intrinsic tryptophan fluorescence and scans from 350 nm and 330 nm were recorded as a function of temperature to monitor changes upon thermal unfolding. The resulting unfolding profile curves are automatically analyzed for inflection temperatures (Ti) based on the brightness ratio 350 nm/330 nm is used to determine the initial ratio (at 35°C) and $\Delta$ ratio (the difference between the initial ratio at 35°C and final ratio at 95°C). Data were analyzed using GraphPad Prism 9 and normalized to the WT Ycf1 melting temperature (50.8°C). Three biological replicates were performed in total for each sample.

### LC-MS/MS analysis of phosphorylated Ycf1

LC-MS/MS analysis was used to identify the phosphorylated residues of Ycf1. LC-MS/MS was performed at the University of Arizona mass spectrometry core using a Thermo Fisher Scientific Q Exactive Plus. Separations were performed on an Acclaim PepMap RSLC column (75 $\mu$m × 25 cm) using a gradient of solvent A (water and 0.1% formic acid) and solvent B (acetonitrile, 0.1% formic acid). The gradient uses 3–20% solvent B over 90 min, 20–50% solvent B over 10 min, 50–95% solvent B for 10 min, and finally a 95% solvent B for 10 min with a final 3% solvent B for 10 min. The Xcalibur v 4.0.27.19 software (68) was used for data-dependent acquisition with a 70,000 resolution. A range of 350–1,500 mass/charge (m/z) with automatic gain control set to 1 × 10$^6$ and a maximum injection time (IT) of 65msec was used and the 10 most intense ions were subjected to higher energy collisional dissociation (HCD) at 27 normalized collision energy with 1.5 m/z, AGC of 5e4, and maximum IT of 65 msec and dynamic exclusion used to exclude single MS/MS ions for 30 s following an acquisition. Ions with charge states of +1, >7, unassigned, or isotopes are excluded.

MS and MS/MS data were analyzed against the *S. cerevisiae* UniProt database containing 9,124 sequence entries, using Thermo Proteome Discoverer v 2.2.0388 with additional common contaminant proteins added. Tryptic peptides were identified and considered matches with up to two missed cleavage sites and variable modifications considered include phosphorylation (98.00 D), methionine oxidization (15.995 D) and cysteine carbamidomethylation (57.021 D). Protein identification was performed at 99% confidence with XCorr (69) in a reversed database search and label-free quantification was also performed. Protein and peptide data were further analyzed using Scaffold Q+S v4.8.7. (70). Protein identifications will be accepted that pass a minimum of two peptides identified at 0.1% peptide false discovery rate and

90–99.9% protein confidence by the Protein Profit algorithm within Scaffold.

### Yeast spotting assay

Yeast survival assays were performed with BY4741—YDR135C Ycf1 knockout *S. cerevisiae* strain as reported in reference 71. Cells were chemically transformed (Zymogen Inc.) with WT Ycf1, E1435Q Ycf1 and R-domain scrambled plasmids. Transformants were plated onto YNB-His plates and incubated for 2–3 d at 30°C. Once colonies had developed, individual colonies were inoculated into 5 ml YNB-His liquid media and grown overnight at 30°C when shaking at 200 rpm. Yeast cells were pelleted by slow centrifugation at 800*g* for 5 min, then washed with 0.9% saline solution before pelleting them again. Final resuspension was carried out in 1–3 ml sterile water; absorbance at OD600 was read for all cultures. All cultures were then adjusted to 0.5 OD600 (±5% error). Serial dilutions of each culture were performed in a 96-well plate, 200 μl of normalized yeast solution was added to the first row (row A), while the following 3–4 columns were used as technical replicas and 120 μl of sterile water was added to rows B to F. Serial dilution on a 1:5 ratio was performed using 30 μl of row A culture to the following rows, creating a OD600 serial range of 0.5, 0.1, 0.02, 0.004, 0.0008, 0.00016. YRG agar plates (yeast nitrogen base with ammonium sulfate 0.67% wt/vol, raffinose 1% wt/vol, galactose 2% wt/vol, CSM-His 0.077% wt/vol, and 2% wt/vol agar) with and without 100 $\mu$M CdCl$_2$ were shortly incubated at 30°C at until stamping. Cells were then stamped onto 30°C warm YRG plates containing or not 150 $\mu$M CdCl$_2$. Cellularly stamped YRG plates were kept at 30°C for 4 d before imaging on a Bio-Rad ChemiDoc MP Imaging System. Four biological replicates were performed in total for each sample.

### Evolutionary coupling analysis of Ycf1

Evolutionary coupling analysis was performed using the EV couplings web server (https://v2.evcouplings.org) (44) to identify co-evolving residues within the Ycf1 protein (UniProt ID: P39109). The residues K268 to S965 were selected as the search range for the coupling analysis so coverage of the full R-domain region (855–935) could occur. The multiple sequence alignment of the homologous sequence was selected by using the server's default parameters, selecting sequences with a maximum of 80% identity to cover a broad evolutionary spectrum when minimizing redundancy. The co-evolving pairs were then mapped onto the Ycf1 protein structure presented here to identify potential interaction networks and functional sites critical to the R-domain and its phosphorylation sites. Scores were used with a cutoff of ≥1 and a probability of ≥70% to account for the low conservation of the R-domain sequence.

## Data Availability

All supporting data and materials are made available upon request. The Ycf1 structure has been deposited in the Protein Data Bank (PDB) with accession code 9AYC, and the EM data are available in the EMDB with accession code EMD-43985. The UniRef90 database

used in EV coupling analysis can be found at https://www.uniprot.org/help/uniref.

## Supplementary Information

## Acknowledgements

This research received generous support through grants from the National Institute of Allergy and Infectious Diseases (NIAID) under grant number NIH R01 AI156270 (awarded to TM Tomasiak) which supported the Life Sciences North Imaging Facility at the University of Arizona. The study is also supported by the NIH T32 GM008404 grant awarded to MSI Rasel. We acknowledge the support of the National NIH-Funded Pacific Northwest Center for Cryo-EM (PNCC) at the Oregon Health and Science University (OHSU) for funding a portion of this study under the grant U24GM129547 through the proposal ID #160065. Additional funding was provided by the University of Arizona through a BIO5 Postdoctoral Fellowship Award granted to NK Khandelwal. We thank Tarjani Thaker for her assistance in designing the scrambled constructs. We extend our gratitude to the team at the PNCC at OHSU, with special thanks to Nancy Meyer and Claudia Lopez, for their invaluable assistance with data collection. We express our gratitude to the Analytical and Biological Mass Spectrometry Facility at the University of Arizona, especially to Krishna Parsawar and Cynthia David for their work on mass spectrometry-based phosphorylation analysis.

### Author Contributions

R Souza Amado de Carvalho: conceptualization, resources, data curation, software, formal analysis, validation, investigation, visualization, methodology, project administration, and writing—original draft, review, and editing.
MSI Rasel: conceptualization, resources, data curation, software, formal analysis, validation, investigation, visualization, methodology, project administration, and writing—original draft, review, and editing.
NK Khandelwal: conceptualization and methodology.
TM Tomasiak: conceptualization, software, supervision, funding acquisition, validation, project administration, and writing—original draft, review, and editing.

### Conflict of Interest Statement

The authors declare that they have no conflict of interest.

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
