## [Reviewer comments · Life Science Alliance]

Life Science Alliance

Cryo-EM reveals a phosphorylated R-domain envelops the NBD1 catalytic domain in an ABC transporter

Rodolpho Souza Amado de Carvalho, Md Shamiul Islam Rasel, Nitesh Khandelwal, and Thomas Tomasiak

DOI: <https://doi.org/10.26508/lsa.202402779>

Corresponding author(s): Thomas Tomasiak, University of Arizona

Review Timeline:

Submission Date:	2024-04-18
Editorial Decision:	2024-06-06
Revision Received:	2024-07-31
Editorial Decision:	2024-08-01
Revision Received:	2024-08-05
Accepted:	2024-08-06

Transaction Report:

June 6, 2024

Re: Life Science Alliance manuscript #LSA-2024-02779-T

Dr. Thomas M. Tomasiak
University of Arizona
Chemistry and Biochemistry
1041 E Lowel Street
Tucson, Arizona 85719

Dear Dr. Tomasiak,

Thank you for submitting your manuscript entitled "Cryo-EM reveals a phosphorylated R-domain envelops the NBD1 catalytic domain in an ABC transporter" to Life Science Alliance. The manuscript was assessed by expert reviewers, whose comments are appended to this letter. We invite you to submit a revised manuscript addressing the Reviewer comments.

Thank you for this interesting contribution to Life Science Alliance. We are looking forward to receiving your revised manuscript.

Sincerely,

B. MANUSCRIPT ORGANIZATION AND FORMATTING:

Reviewer #1 (Comments to the Authors (Required)):

This manuscript by Carvalho et al presents a new structure of a quad-phosphorylated state structure of the yeast ABC transporter Ycf1 by cryo-EM at 3.2 Angstrom Resolution, identifying four phosphorylated residues in the (R) domain. The authors go on to map out the full R-Domain for the first time, identifying key NBD interactions. The structural data is solid and complemented by ATPase assays, nano differential scanning fluorimetry for thermostability assays, site-directed mutagenesis to generate mutants, and mass spectrometry to analyze the phosphorylated sites. They also used size exclusion chromatography and SDS-PAGE to confirm protein expression, Western blotting to confirm phosphorylation, and yeast spotting assays to evaluate cell viability in different R-domain scrambled plasmids. Insights gleaned from the Ycf1 structure-function data presented here have implications for other members of the ABCC family and, as such, the findings are significant and worthy of publication after the following concerns are addressed:

1. The authors should include a figure with domain /TM helix-based EM density and an EM map colored by local resolution. The density for the R domain appears significantly noisier and at lower resolution, and this would aid in lending further credence to specific residue assignment when talking about R-domain/NBD contacts, specifically in figure EV3A.
2. The Nano-DSF thermostability measurements exhibit high error bars and uncertainty regarding the thermal stability analysis of the PKA-treated sample, as discussed earlier. As a result, the functional aspects of phosphorylation are heavily dependent on the ATPase assay alone. It is also difficult to connect the physiological relevance of phosphorylation on protein stability.

Minor:

3. In the section 'The structure reveals an R-domain with four phosphorylation sites'

Line 2 - contains a typo. 'T914' should be corrected to 'S914'.

4. Figure 3 - Legend

Line 9 - It is stated as 'R803 and H805'. Please check residue assignments - Based on the illustration E, it should be 'H803 and K805'.

5. Appendix: Please check number of catalytic EtoQ mutant as several different residue numbers are used (1455, 1435, and 1432).

Reviewer #2 (Comments to the Authors (Required)):

The paper by de Carvalho and colleagues addresses a poorly understood area of eukaryotic ABC transporter structural biology, namely the structure of the R-domain and its interaction with the rest of the protein (primarily NBD1). By achieving purification of a 4-site phosphorylated YCF1 the authors manage to resolve by cryo-EM the entire R domain and provide a structural and functional analysis of its NBD1 interaction surface. The impact of this work is evident to ABC transporter researchers, particularly those with a focus on ABCC transporters.

I am not conversant with the details of the EM procedure so will reserve comment on any technicality there but the density for the 4 phosphates looks unambiguous and the structural changes accompanying phosphorylation are also evident. The interaction surface is intriguing as it seems to be driven by electrostatic interactions that only become relevant upon phosphorylation and which then drive a change in conformation of the R domain that presumably regulates NBD1 (and hence the entire transporter).

My comments on the paper for improvement are (PROBABLY) really minor as the functional work, the purification and ATPase activity and the MS all look to be unambiguous, correctly analysed and presented and convincing.

Some minor points:

Intro 2nd paragraph should be clearly that it is eukaryotic ABCs that are classified into 7 subfamilies.

1st paragraph of results penultimate line thereof is awkward; I think it should say that the ATPase activity of the dephosphorylated protein is reduced. At the moment it doesn't say that.

POSSIBLE MAJOR REVISION: At least 2 figure legends (Figure 4 and Figure 1) both refer to "n=6 (is technical replicates)". As I am sure the authors know, technical replicates are not the same as "n". Can we get some clarity that where they say n=6 it is 6 experiments, not one experiment with 6 repeat pipettings. If the functional work DOES represent a single experiment then this work needs to be repeated (Figure 1E, Figure 2A, Figure 4A)

Inspection of Figure EV4 made me wonder if co-evolution analysis showed connections between 785-803 and 884-900 given their proposed contact interact?

Figure EV5 panel B has some wrongly aligned connecting loops, particularly towards the C-terminus of that figure.

In the methods there is at least one mention of a filter size by Molarity rather than by micrometers.

In the methods the ATPase assay appears to be done at a ridiculous scale (mls) rather than microlitres. The ATPase assay also refers to transport which it should not do as no transport assays are shown. Again this section contains an "n=4 technical replicates" statement which is puzzling.

Reviewer #3 (Comments to the Authors (Required)):

The manuscript from de Carvalho and colleagues is investigating the structural and functional role of specific phosphorylated residues to activate the Ycf1 transporter. Ycf1 belongs to the ABCC family that is regulated by phosphorylation of the regulatory R-domain. Several biochemical studies and structures have revealed the interplay of phosphorylation and conformational changes associated with activation of the ABCC family. Here, they present the cryo-EM structure of Ycf1 in a fully phosphorylated state that has the entire phosphorylated R-domain resolved where it was partially resolved in their previous work. They further investigate the role of phosphorylated residues by mutagenesis and cell based survival assays.

The design of the functional work is of interest/novel, especially the scrambled chimeras as it allows for a 'fast' way to evaluate the role of phosphorylation on specific residues without disrupting the length of the R-domain.

Although the structural work does not bring any novel or new understanding on the ABCC family, the new model can be valuable from a mechanistic perspective and especially on the stabilisation of the phosphorylated R-domain. The authors may want to add in their discussion that these interactions possibly stabilise the phosphorylated R-domain out of the TMD rather than flopping around during the transport cycle. The interactions with the NBD1 are unlikely to drive transport as the NBDs are still separated; it probably provides a scaffold for the R-domain to be stabilised while the transporter undergoes conformational changes towards the outward facing conformation upon ATP binding, similar to the outward facing CFTR (<https://doi.org/10.1016/j.cell.2017.06.041>).

I am not so sure about the statement 'The phosphorylated residue (S903) shown here directly interacts with a highly basic pocket in the lower NBD1 portion and drives Ycf1 function in our cellular assay'. The cell toxicity data show that phosphorylation of Ser903 does drive the function but the interactions are not a requirement. The recent ABCC2 work (<https://www.nature.com/articles/s41467-024-46392-8>) shows that the equivalent residue also drives transport activity by displacing the R-domain. An interesting experiment to validate if the interactions do drive transport would be to replace Ser903 with a bulky aa such as Phe or Trp and measure the cell killing (not necessary for this work).

In addition: 'a new 4th phosphorylated site, S903, ' is not quite true and it should be tuned down as they have previously shown to be important for the function of Ycf1.

Minor correction:

Appendix Table S2: the B-factor is reported as 11777, shouldn't it be 117.77?

Life Science Alliance Manuscript
Editorial Decision LSA-2024-02779-T
Reviewers' Comments Addressed

Reviewer #1Comment 1

"The authors should include a figure with domain /TM helix-based EM density and an EM map colored by local resolution. The density for the R domain appears significantly noisier and at lower resolution, and this would aid in lending further credence to specific residue assignment when talking about R-domain/NBD contacts, specifically in figure EV3A."

Authors:

We thank the reviewer for this suggestion and have added a new figure to show this information (Figure S4. **Cryo-EM map quality and domain features of the tetra-phosphorylated Ycf1 state**). We present a local resolution map of the tetra-phosphorylated Ycf1 and R-domain demonstrating the resolution quality in observance of the residue assignment in a more detailed manner. We agree with the reviewer that this density is weak in many places and indeed is the impetus for our systemic functional investigation of these regions with our scrambled construct mutants.

Comment 2

"The Nano-DSF thermostability measurements exhibit high error bars and uncertainty regarding the thermal stability analysis of the PKA-treated sample, as discussed earlier. As a result, the functional aspects of phosphorylation are heavily dependent on the ATPase assay alone. It is also difficult to connect the physiological relevance of phosphorylation on protein stability."

Authors:

We thank the reviewer and have reperfomed protein purification of the different Ycf1 samples again and reperfomed the Nano-DSF thermostability using three biological replicates. The updated data shows low error bars and suggests that there is little physiological difference between PKA-treated and wild-type, but that there is a 5°C difference between PKA-treated and dephosphorylated. Additionally, we performed a one-way ANOVA and Dunnett's test for the three biological replicates, which shows the values are significantly different. We have updated the figure and modified the text to reflect these differences. We infer that any positive allosteric effect that the phosphorylated R-domain has on NBD1 is therefore likely to be small.

Comment 3

"In the section 'The structure reveals an R-domain with four phosphorylation sites' Line 2 - contains a typo. 'T914' should be corrected to 'S914'."

Authors:

We thank the reviewer and have fixed the typo.

Comment 4

"Figure 3 - Legend Line 9 - It is stated as 'R803 and H805'. Please check residue assignments - Based on the illustration E, it should be 'H803 and K805'."

Authors:

We have fixed the typo 'R803 and H805' to 'H803 and K805' as presented in the structure.

Comment 5

"Appendix: Please check number of catalytic EtoQ mutant as several different residue numbers are used (1455, 1435, and 1432)."

Authors:

We thank the reviewer and after revision, we have corrected the numbering for the catalytic EtoQ mutant to its correct form (E1435Q) in Figure S7 as well as in the Materials and Methods, Yeast Spotting Assay section.

Reviewer #2

Comment 1

"Intro 2nd paragraph should be clearly that it is eukaryotic ABCs that are classified into 7 subfamilies"

Authors:

We thank the reviewer and have complemented the statement including that eukaryotic ABC proteins are classified in 7 subfamilies, with 5 of them transporters.

The statement has been modified in the Introduction, 2nd paragraph, line 1 to:

"Eukaryotic ABC proteins are classified into seven subfamilies (A to G) based on their topology and domain packing."

Comment 2

"1st paragraph of results penultimate line thereof is awkward; I think it should say that the ATPase activity of the dephosphorylated protein is reduced. At the moment it doesn't say that."

Authors:

We thank the reviewer and after revision, we have reformulated the sentence for clarity stating that the ATPase activity of the dephosphorylated protein is reduced.

The sentence in the Results section, 1st paragraph, line 20 has been replaced to:

"The ATPase activity assay with different phospho-states of Ycf1 has shown that upon PKA treatment the ATP hydrolysis rate increases while the dephosphorylated protein is reduced compared to the wild-type."

Comment 3

"POSSIBLE MAJOR REVISION: At least 2 figure legends (Figure 4 and Figure 1) both refer to "n=6 (is technical replicates)". As I am sure the authors know, technical replicates are not the same as "n". Can we get some clarity that where they say n=6 it is 6 experiments, not one experiment with 6 repeat pipetting's. If the functional work DOES represent a single experiment then this work needs to be repeated (Figure 1E, Figure 2A, Figure 4A)"

Authors:

We thank the reviewer and have reperfomed this analysis. We have repeated the experiment in Figure 2A, now utilizing three biological replicates each with two technical replicates instead of the original experiment, which were only technical replicates as the reviewer mentioned. The figure and legend have been updated to reflect this change. Similar ATPase activity has been observed for these experiments compared to the previous one except for a small increase in the phosphorylated sample activity. We infer that this small variation is physiologically nearly identical to the original sample and our conclusions do not change. We believe that this has increased the rigor of our biochemical results. Similarly, we also reperfomed the thermostability measurements in Figure 1E with three biological replicates.

For Figure 4A, the experiment was not repeated as it already represented one of the four biological replicate plates and we have corrected the legend removing the incorrect description of n=6 technical replicates to four biological replicates (Source data has been attached that represents the replicates).

Comment 4

"Inspection of Figure EV4 made me wonder if co-evolution analysis showed connections between 785-803 and 884-900 given their proposed contact interact?"

Authors:

We thank the reviewer and have reviewed the co-evolution analysis and indeed there are no connections between 785-803 and 884-900.

Comment 5

"Figure EV5 panel B has some wrongly aligned connecting loops, particularly towards the C-terminus of that figure."

Authors:

We thank the reviewer and have fixed the alignment of the connecting lines in the figure.

Comment 6

"In the methods there is at least one mention of a filter size by Molarity rather than by micrometers."

Authors:

We thank the reviewer and have corrected the typo regarding the filter dimensions to micrometers (μm).

The sentence has been rewritten in Materials and Methods, 2nd paragraph, line 10 as:

"The clarified supernatant was filtered through a 0.4 μm filter to remove the insoluble fraction and supplemented with 30 mM Imidazole pH 7.0 immediately before loading at a flow rate of 2 mL/min onto a 5 mL Ni-NTA immobilized metal affinity chromatography (IMAC) column (Bio-Rad) equilibrated in Buffer A (50 mM Tris-Cl, 300 mM NaCl, 0.01% LMNG/0.001% CHS, pH 7.0)"

Comment 7

"In the methods the ATPase assay appears to be done at a ridiculous scale (mls) rather than microlitres.

The ATPase assay also refers to transport which it should not do as no transport assays are shown. Again this section contains an "n=4 technical replicates" statement which is puzzling."

Authors:

We thank the reviewer and have corrected the typos regarding the volume scale, replacing with the correct microliter (μL) scale used in the experimental steps.

The sentence has been rewritten in Materials and Methods, 8th paragraph, line 11 as:

"The ATPase reaction was started by the addition of an Mg-ATP solution prepared in Tris buffer, bringing the final concentration of Ycf1 to 0.135 mM in a 50 μL reaction containing 10 mM MgCl_2 and varying final concentrations of ATP (0–4 mM). Samples in the absence of Mg-ATP solution or Ycf1 were prepared as negative controls. The reaction was quenched for a single time point by adding 40 μL of 5% SDS. Following quenching, 200 μL of detection reagent (8.75 mM Ammonium Molybdate, 3.75 mM Zinc Acetate, pH 5.0, and 7.5% Ascorbic Acid pH 5.0 prepared fresh prior to use) were added to each sample and incubated for 25 minutes at 37°C for revelation."

Additionally, we have fixed the word "transport" to "ATPase rates". As this experiment was repeated, the correct biological and technical replicates are now corrected as described in Reviewer's comment 3.

The sentence has been rewritten in Materials and Methods, 8th paragraph, line 5 as:

"The ATPase rates were measured as a colorimetric endpoint assay for inorganic phosphate detection described by (61), which allowed the indirect quantification of protein-driven ATP hydrolysis. The Michaelis-Menten kinetics for the ATPase rates of three biological replicates at an ATP range (0.05 to 4 mM) are reported as the mean \pm standard quantified in GraphPad Prism. For each biological replicate, two technical replicates were performed."

Reviewer #3

Comment 1

"Although the structural work does not bring any novel or new understanding on the ABCC family, the new model can be valuable from a mechanistic perspective and especially on the stabilization of the phosphorylated R-domain. The authors may want to add in their discussion that these interactions possibly stabilize the phosphorylated R-domain out of the TMD rather than flopping around during the transport cycle. The interactions with the NBD1 are unlikely to drive transport as the NBDs are still separated; it probably provides a scaffold for the R-domain to be stabilized while the transporter undergoes conformational changes towards the outward facing conformation upon ATP binding, similar to the outward facing CFTR (<https://doi.org/10.1016/j.cell.2017.06.041><<https://doi.org/10.1016/j.cell.2017.06.041>>)."

Authors:

We thank the reviewer for this suggestion and have added a new last sentence in the third paragraph of the discussion that states this idea. Indeed, our previously published dephosphorylated model suggests that disruption of the negative orthostatic and/or competitive effect of the R-domain on substrate binding is the dominant effect on activation. This effect has also been shown to be the case in CFTR and MRP2.

We have added in the Discussion section, 3rd paragraph, line 13:

"These interactions likely stabilize the phosphorylated R-domain outside of the TMD to sequester it during the transport cycle. The recent structures of human Mrp2 with different conformational states suggest a similar mechanism in which the auto-inhibited state must be disengaged upon substrate binding"

Comment 2

"I am not so sure about the statement 'The phosphorylated residue (S903) shown here directly interacts with a highly basic pocket in the lower NBD1 portion and drives Ycf1 function in our cellular assay'. The cell toxicity data show that phosphorylation of Ser903 does drive the function but the interactions are not a requirement. The recent ABCC2 work (<https://www.nature.com/articles/s41467-024-46392-8>) shows that the equivalent residue also drives transport activity by displacing the R-domain. An interesting experiment to validate if the interactions do drive transport would be to replace Ser903 with a bulky aa such as Phe or Trp and measure the cell killing (not necessary for this work)."

Authors:

We thank the reviewer and have modified the sentence mentioned in the Discussion section, 3rd paragraph, line 1 to:

"The phosphorylated residue (S903) shown here can directly interact with a highly basic pocket in the lower NBD1 portion and drive partial Ycf1 function in our cellular assay".

We agree that we do not know if this region has a positive allosteric effect in a permissive function or simply removes the R-domain from between the NBDs in a permissive function. The functional data we show can only infer a possible limited effect directly on NBD1.

Comment 3

"In addition: 'a new 4th phosphorylated site, S903, ' is not quite true and it should be tuned down as they have previously shown to be important for the function of Ycf1."

Authors:

We thank the reviewer and have corrected this sentence in the Discussion section, 1st paragraph, line 1 to: "It describes a 4th phosphorylated site, S903 which has not been structurally observed before".

That way we point to the novelty in the structural observation for that phospho-site.

Comment 4

"Appendix Table S2: the B-factor is reported as 11777, shouldn't it be 117.77?"

Authors:

This number is from phenix and refers to the number of atoms with b-factors refined as isotropic and anisotropic b-factors. The line below it shows the average b-factor over these atoms, which incidentally is similar (~118).

August 1, 2024

RE: Life Science Alliance Manuscript #LSA-2024-02779-TR

Dr. Thomas M. Tomasiak
University of Arizona
Chemistry and Biochemistry
1041 E Lowell Street
Tucson, Arizona 85719

Dear Dr. Tomasiak,

Thank you for submitting your revised manuscript entitled "Cryo-EM reveals a phosphorylated R-domain envelops the NBD1 catalytic domain in an ABC transporter". We would be happy to publish your paper in Life Science Alliance pending final revisions necessary to meet our formatting guidelines.

- please be sure that the authorship listing and order is correct
- please add a Running Title to our system
- please add ORCID ID for the corresponding author -- you should have received instructions on how to do so
- please consult our manuscript preparation guidelines <https://www.life-science-alliance.org/manuscript-prep> and make sure your manuscript sections are in the correct order (sections such as Data Availability, Acknowledgements, Author contributions, and Conflict of Interest should be placed after the Materials and Methods section and before the references section)
- please add your main, supplementary figure, and table legends to the main manuscript text after the references section
- please exclude figures from the manuscript text and leave them uploaded separately
- there is a call-out for Figure 4C on pg. 6, and this figure has only A and B panels; please correct
- please add call-outs for Figures 1E; S1A-C; S2A-C; S3A-D; S4A-C; S6A-C and S7B-D to your main manuscript text
- please remove the "Significance Statement" provided after the abstract

LSA now encourages authors to provide a 30-60 second video where the study is briefly explained. We will use these videos on social media to promote the published paper and the presenting author (for examples, see <https://docs.google.com/document/d/1-UWCfbE4pGcDdcgzcmiuJl2XMBJnxKYeqRvLLrLSo8s/edit?usp=sharing>). Corresponding or first-authors are welcome to submit the video. Please submit only one video per manuscript. The video can be emailed to contact@life-science-alliance.org

A. FINAL FILES:

B. MANUSCRIPT ORGANIZATION AND FORMATTING:

Sincerely,

August 6, 2024

RE: Life Science Alliance Manuscript #LSA-2024-02779-TRR

Dr. Thomas M. Tomasiak
University of Arizona
Chemistry and Biochemistry
1041 E Lowell Street
Tucson, Arizona 85721

Dear Dr. Tomasiak,

Thank you for submitting your Research Article entitled "Cryo-EM reveals a phosphorylated R-domain envelops the NBD1 catalytic domain in an ABC transporter". It is a pleasure to let you know that your manuscript is now accepted for publication in Life Science Alliance. Congratulations on this interesting work.

DISTRIBUTION OF MATERIALS:

Again, congratulations on a very nice paper. I hope you found the review process to be constructive and are pleased with how the manuscript was handled editorially. We look forward to future exciting submissions from your lab.

Sincerely,
